# Noninvasive Evaluation of Myocardial Work in Patients with Chronic Kidney Disease Using Left Ventricular Pressure-Strain Loop Analysis

**DOI:** 10.3390/diagnostics12040856

**Published:** 2022-03-30

**Authors:** Xiaohua Liu, Lixin Chen, Xiaofang Zhong, Guijuan Peng, Yuanyuan Sheng, Jian Li, Qian Liu, Bobo Shi, Yuxiang Huang, Jinfeng Xu, Yingying Liu

**Affiliations:** Shenzhen Medical Ultrasound Engineering Center, Department of Ultrasound, Shenzhen People’s Hospital (The Second Clinical Medical College, Jinan University, The First Affiliated Hospital, Southern University of Science and Technology), Shenzhen 518020, China; lengfeng0418@163.com (X.L.); neostar84@aliyun.com (L.C.); zxflds1988@163.com (X.Z.); funnypgg2019@163.com (G.P.); shengyyah@163.com (Y.S.); dr_lijian@126.com (J.L.); zctsdzct@163.com (Q.L.); sumi150117@163.com (B.S.); ln4162643454@163.com (Y.H.)

**Keywords:** chronic kidney disease, myocardial work, pressure-strain loop analysis, left ventricular hypertrophy

## Abstract

(1) Objective: To evaluate myocardial injury by observing the different parameters of global myocardial work (MW) by left ventricular pressure-strain loop (PSL) analysis in patients with chronic kidney disease (CKD). (2) Methods: According to the left ventricular mass index, the study patients with CKD were further divided into two groups: the left ventricular normal group (CKD_N-LVH_, 59) and left ventricular hypertrophy group (CKD_LVH_, 46). Thirty-three healthy controls (CON) matched in age and sex with the CKD group were recruited. The routine ultrasonic parameters were obtained by routine TTE, and the strain index and different parameters of the left ventricular MW were obtained by dynamic image offline analysis. (3) Results: This study found that (1) compared with the CON group, the CKD_N-LVH_ group had a significantly increased global waste work (GWW) and significantly decreased global work efficiency (GWE), the GWW further increased, and GWE further decreased in the CKD_LVH_ group. There was no significant change in the global work index (GWI) and global constructive work index (GCW) in the CKD_N-LVH_ group, but the GWI and GCW in the CKD_LVH_ group were significantly increased. (2) According to the grouping analysis of systolic blood pressure (SBP), we found that the GWW increased and GWE decreased in CKD patients with an elevated SBP. (3) Correlation analysis showed that the increase of the peak strain dispersion, SBP, and left ventricular mass index and the decrease of the estimated glomerular filtration rate were significantly correlated with the decrease of the GWE and the increase of the GWW. (4) Receiver operating characteristic curve analysis showed that the area under the curve (AUC) of myocardial damage induced by the GWE and GWW in the CKD group and CON group was higher than that of left ventricular global longitudinal strain (AUCs: 0.87 and 0.878 versus 0.72, respectively). (4) Conclusions: Noninvasive left ventricular PSL analysis can be used to evaluate the global MW in patients with CKD. The study justified the role of GWW in the noninvasive assessment of myocardial function in patients with CKD.

## 1. Introduction

Chronic kidney disease (CKD) is a common disease that seriously affects human life and health. The pathophysiological basis of myocardial injury caused by CKD has been widely studied. CKD is usually associated with left ventricular hypertrophy (LVH) and an increase in the left ventricular mass (LVM), which may increase the risk of cardiovascular disease and death [1,2,3,4]. Hypertension itself is an important risk factor for cardiovascular disease in CKD, and it is almost always present in patients with renal failure [5]. Hypertension also plays a major role in cardiac damage in CKD by inducing LVH [6,7].

The left ventricular (LV) two-dimensional speckle-tracking outweighs LV ejection fraction (EF) in predicting cardiac events [8], but it is affected by load, which may affect the evaluation of the myocardial work (MW) [9]. LV pressure-strain loop (PSL) analysis studies the relationship between the strain and afterload through the strain index and noninvasive LV pressure changes, and provides a new quantitative method of MW measurement. Recent studies have confirmed that the noninvasive MW assessment improves understanding of the relationship between LV remodeling and increased ventricular wall stress under different loads, and it is superior to strain indexes, such as the global longitudinal strain (GLS), for identifying acute coronary artery occlusion in patients with non-ST segment elevation acute coronary syndrome [10,11].

Therefore, in the present study, we aimed (1) to evaluate the MW in patients with CKD by using a new noninvasive PSL analysis method; (2) to further explore the effect of LVH on MW in patients with CKD; (3) to study the independent correlation between the baseline parameters and the MW index; and (4) to compare the MW index with other echocardiographic parameters, especially strain parameters, in terms of myocardial injury estimation.

## 2. Materials and Methods

### 2.1. Ethics Statements

The present study was approved by the Ethics Committee of Shenzhen People’s Hospital and conformed to the standards of the Helsinki Declaration of the World Medical Association. All participants provided written informed consent.

### 2.2. Study Design and Population

The study population included 105 patients with CKD who were treated at Shenzhen People’s Hospital from November 2020 to March 2021. CKD was diagnosed based on the following diagnostic criteria [12]: glomerular filtration rate (GFR) < 60 mL/min/1.73 m^2^ for ≥3 months, with or without kidney injury, including kidney structural and functional abnormalities for ≥3 months, with or without a decrease in the GFR. Thirty-three healthy subjects were selected as the control (CON) group. Patients with suboptimal image quality for myocardial deformation analysis, low EF (<50%), coronary heart disease, severe valvular disease, congenital heart disease, severe arrhythmia, and renal transplantation were excluded. All patients were clinically and hemodynamically stable.

### 2.3. Clinical Features

The following clinical characteristics of included subjects were collected: sex, age, height, weight, blood pressure, heart rate, estimated GFR (eGFR), serum creatinine (SCR) level, blood urea nitrogen (BUN) level, echocardiographic parameters, body mass index (BMI), BMI = weight (kg)/height^2^ (m^2^), and body surface area (BSA), BSA = (Height (cm) × Weight (kg)/3600)^½^.

### 2.4. Echocardiographic Analysis

For echocardiographic analysis, we used the GE Vivid E95 ultrasonic diagnostic instrument equipped with an M5S probe (3.5 MHz). The patient was asked to take the left recumbent position; we connected the electrodes to the patient and recorded electrocardiography synchronously; then a comprehensive transthoracic echocardiography (TTE) examination was conducted with the patient in the resting state to obtain the best image quality. The LV EF (LVEF) and LV end-diastolic volume (LVEDV) were determined by the biplane Simpson method. M-mode echocardiography was performed from the parasternal long-axis section to measure the LV internal diameter at end-diastole (LVIDd), LV internal diameter at end-systole, interventricular septum diameter at end-diastole (IVSd), and posterior wall thickness at end-diastole (PWTd). According to the Devereux correction formula, the LVM, LVM (g) = 0.8 × {1.04 × [(IVSd + PWTd + LVIDd)^3^ − LVIDd^3^] + 0.6, and LV mass index (LVMI), LVMI (g/m^2^) = LVM/BSA, were calculated as the criteria for evaluating LVH (LVMI ≥ 115 g/m^2^ in men and LVMI ≥ 95 g/m^2^ in women) [13]. LVEDVI was standardized according to BSA (LVEDV/BSA). All records and measurements were performed in accordance with the guiding principles of the American Society of Echocardiography [14].

### 2.5. Left Ventricular Global Longitudinal Strain and Myocardial Motion Synchronization

Continuous dynamic images of the LV apical four-chamber, three-chamber, and two-chamber views (50–80 frames/s) were obtained. The images were imported into the software (Echo PAC version 203 GE; Vingmed Ultrasound) in the format of stored digital raw data for analysis. Automatic functional imaging was used to automatically track each endocardial and epicardial boundary in the three apical dynamic images, and the region of interest was adjusted by correcting the endocardial boundary or width if necessary. The average value of the peak longitudinal strain of 17 segments of the LV myocardium was determined as the myocardial GLS (an absolute value). At the same time, the peak strain dispersion (PSD) value was obtained to evaluate the synchronization of the myocardial contraction in the left ventricle.

### 2.6. Quantitative Analysis of Myocardial Work Done by the Left Ventricle

By combining the LV pressure curve of the LV strain and a noninvasive estimation, the MW was calculated using GE software (EchoPAC^TM^ version 203, GE-Vingmed Ultrasound AS, Horten, Norway). Before echocardiography, the brachial artery pressure was measured using a cuff manometer three times, and the average value was used in the analysis. It was assumed that the peak systolic pressure of the left ventricle is equal to the brachial artery pressure. According to the definition of the isovolumic contraction period and ejection period duration (between valve opening and closing), the software creates a noninvasive LV pressure curve [15]. The area in the PSL curve represents the global work index (GWI) of the myocardium. Through the analysis using MW software, the following parameters were obtained.

GWI (mmHg%): the total work done in the LV PSL analysis calculated from mitral valve closure to mitral valve opening (except in diastole).Global constructive work (GCW, mmHg%): contributes to LV ejection work, including systolic myocardial shortening and isovolumic diastolic myocardial elongation.Global waste work (GWW, mmHg%): not conducted for LV ejection work, including systolic myocardial elongation and isovolumic diastolic period shortening and increasing.Global work efficiency (GWE, %): (GCW/(GCW + GWW) × 100%.

### 2.7. Statistical Analysis

The normal distribution of data was checked using Shapiro–Wilk test. Continuous variables are presented as $\bar{x}$ ± s or median (quartile range). When the variance was homogeneous and followed a normal distribution, single-factor analysis of variance was used to perform comparisons among the groups and pairwise comparisons between the groups. The least significant difference t-test was used when the variance was normally distributed; Kruskal–Wallis test was used when the variance was non-normally distributed; and the chi-squared test was used to compare counting data between the groups. Pearson’s correlation coefficients were used to evaluate correlations between different variables; Spearman correlation coefficients were used when the distributions were non-normal. The area under the receiver operating characteristic (ROC) curve was used to compare the accuracy of the GLS and MW parameters in the identification of myocardial injury. Inter-observer agreement was assessed by two independent investigators who randomly selected 15 subjects. Intra-observer consistency was assessed by the same investigator in two analyses of the 15 subjects. Intraclass correlation coefficients (ICCs) and the Bland-Altman method were used to evaluate the consistency of MW parameters between and within the observers. SPSS statistical software version 26.0 (IBM Corp., Armonk, NY, USA) was used to analyze the data. All tests were two-sided, and *p* < 0.05 was considered to be statistically significant.

## 3. Results

### 3.1. Clinical and Laboratory Characteristics

In total, 138 subjects were enrolled in this study, including 105 patients with CKD (CKD_N-LVH_ group, 59; CKD_LVH_ group, 46) and 33 healthy participants (CON group). The average age of patients with CKD (61 men and 44 women) was 53.7 ± 15.7 years. Table 1 presents the clinical and laboratory characteristics of each group. There was no significant difference in age, sex, BMI, and BSA between the groups (*p* > 0.05). Compared with the CON group, the CKD group had a higher SBP and diastolic blood pressure, faster heart rate, lower eGFR, and higher SCR and BUN (all, *p* < 0.05).

### 3.2. Transthoracic Echocardiographic Parameters

Table 2 summarizes the TTE parameters. The LVEF of the CKD_LVH_ group was decreased compared with that of the other two groups, whereas the IVSd, LVIDd, LVPWTd, LVEDV, LVEDVI, and LVMI were significantly increased (*p* < 0.05).

### 3.3. Left Ventricular Global Longitudinal Strain and Myocardial Asynchrony

Compared with the CON group, the CKD group had a decreased GLS, and the GLS further decreased in the CKD_LVH_ group compared with the CKD_N-LVH_ group. The PSD increased significantly in the CKD group compared with the CON group, and it further increased in the CKD_LVH_ group compared with the CKD_N-LVH_ group (*p* < 0.05). According to the subdivision analysis of the SBP between the CON group and CKD group, the GLS of the CKD group was lower than that of the CON group (*p* < 0.05), and the GLS of patients with CKD and SBP ≥ 140 mmHg was slightly lower than that of patients with CKD and SBP < 140 mmHg, although the difference was no significant (*p* > 0.05). The PSD was higher in the CKD group than in the CON group. The PSD of patients with CKD and SBP ≥ 140 mmHg was higher than that of patients with CKD and SBP < 140 mmHg (*p* < 0.05). Table 3 and Table 4 show the detailed data.

### 3.4. Myocardial Work Analysis

Compared with the CON group, the CKD group had a decreased GWE and increased GWW, whereas the GWW further increased in the CKD_LVH_ group (*p* < 0.05). The GWI and GCW were higher in the CKD_LVH_ group than in the CKD_N-LVH_ and CON groups (both *p* < 0.05) (Figure 1A–D). According to the subdivision analysis of the SBP between the CON group and CKD group, the GWE of patients with CKD decreased, whereas the GWI, GCW, and GWW of patients with CKD increased; additionally, the GWI, GCW, and GWW of patients with CKD and SBP ≥ 140 mmHg were further increased (*p* < 0.05) (Figure 2A–D). Table 3 and Table 4 show the detailed data.

### 3.5. Independent Correlation Analysis of Various Parameters of Myocardial Work in Patients with Chronic Kidney Disease

Univariate correlation analysis showed that the GWE was positively correlated with the GLS and eGFR (r = 0.42, *p* < 0.05 and r = 0.44, *p* < 0.05, respectively) and negatively correlated with the PSD and LVMI (r = −0.69, *p* < 0.05 and r = −0.33, *p* < 0.05, respectively); GWI was positively correlated with the SBP, GLS, and LVMI (r = 0.67, *p* < 0.05; r = 0.35, *p* < 0.05; and r = 0.31, *p* < 0.05, respectively); GCW was positively correlated with the SBP, GLS, and LVMI (r = 0.72, *p* < 0.05; r = 0.37, *p* < 0.05; and r = 0.30, *p* < 0.05, respectively); and GWW was positively correlated with the SBP, PSD, and LVMI (r = 0.50, *p* < 0.05; r = 0.70, *p* < 0.05; and r = 0.41, *p* < 0.05, respectively) and negatively correlated with the GLS and eGFR (r = −0.29, *p* < 0.05 and r = −0.46, *p* < 0.05, respectively).

### 3.6. Accuracy of Identification of Myocardial Injury between Patients with Chronic Kidney Disease and the Controls

There were significant differences in the GWE, GWW, and GLS between the CKD and CON groups, and the areas under the curves were 0.87, 0.878, and 0.72, respectively (Figure 3). The accuracy of the GWE and GWW in the diagnosis of myocardial injury in patients with CKD was better than that of the GLS. ROC analysis of the GWE showed that the best cutoff value was ≤94%, with a specificity of 93.9% and sensitivity of 68.6%. ROC analysis of the GWW showed that the best cutoff value was >111 mmHg%, with a specificity of 93.9% and sensitivity of 68.6%.

### 3.7. Repeatability of the Parameters of Myocardial Work

Intra-observer and inter-observer consistencies were assessed for each parameter of MW. The intra-observer ICC values of the GWE, GWI, GCW, and GWW were 0.939 (95% confidence interval (CI): 0.829–0.979), 0.78 (95% CI: 0.462–0.92), 0.923 (95% CI: 0.786–0.973), and 0.949 (95% CI: 0.855–0.982), respectively (Figure 4A–D). The inter-observer ICC values were 0.919 (95% CI: 0.776–0.972), 0.952 (95% CI: 0.863–0.982), 0.898 (95% CI: 0.724–0.965), and 0.811 (95% CI: 0.525–0.932), respectively (Figure 4E–H). The ICC values of all parameters were >0.75, indicating good reproducibility.

## 4. Discussion

### 4.1. Main Findings

This study mainly describes the noninvasive evaluation of the MW in patients with CKD by LV PSL analysis. The main findings are as follows: (1) compared with the CON group, the CKD_N-LVH_ group did not have abnormal routine ultrasonic parameters, but the work-related parameters changed—the GWW significantly increased and the GWE decreased; (2) GWW increased and GWE decreased in the CKD_LVH_ group compared with the CKD_N-LVH_ group; moreover, compared with the CON and CKD_N-LVH_ groups, the CKD_LVH_ group had an increased GWI and GCW; (3) GWW increased and GWE decreased in patients with CKD and normal SBP. In patients with CKD and elevated SBP, GWW further increased and GWE further decreased, but there was no significant difference in the strain index between the SBP subgroups; (4) the increase of the PSD, SBP, and LVMI and the decrease of the eGFR were significantly correlated with the decrease of the GWE and the increase of the GWW; (5) the accuracy of the GWE and GWW in distinguishing myocardial injury between the CKD group and CON group was higher than that of the GLS.

### 4.2. Influence of Left Ventricular Hypertrophy on Myocardial Work

LVH is common in CKD, and the incidence of LVH increases with the progressive decline of renal function [16]. In the present study, the GWW increased in the CKD group, resulting in a decrease in the GWE, especially in the CKD_LVH_ group: GCW and GWW increased, and GWE further decreased in patients with CKD and LVH. This shows that in patients with LVH, although the MW is compensated, the work efficiency is decreased. GCW was not significantly increased in the CKD_N-LVH_ group, while GWW was significantly increased, indicating that the function of the myocardium was impaired when there was no obvious structural abnormality, such as hypertrophy and remodeling. In the CKD_LVH_ group, there was a compensatory increase of the GCW due to cardiac hypertrophy, but because hypertrophic myocardial function damage is more severe [17], the GWW increased more significantly, and the GWE decreased. The increase of GWW is key to this change, which may be due to the non-synchronization of LV myocardial motion in patients with CKD. This has been confirmed in previous studies in which LV asynchrony in patients with CKD significantly affected cardiac electrophysiology, systolic function, and regional myocardial perfusion and decreased MW efficiency [15,18,19]. Some studies have found that the GWW increases in different myocardial pacing groups, but with cardiac resynchronization therapy, the GWW decreased significantly [20]. In the correlation analysis of the current study, we found that the GWW is independently related to the PSD. The data of our study also showed an increase of the PSD in patients with CKD, which was consistent with the findings of previous studies [18,21,22], indicating that there was myocardial asynchrony in patients with CKD, and with the progression of myocardial remodeling, the situation of myocardial asynchrony in the CKD_LVH_ group was further aggravated, which led to the increase of the GWW and then the decrease of work efficiency.

### 4.3. Influence of the Systolic Blood Pressure on Myocardial Work

Considering the effect of SBP on MW [10], we further grouped patients with CKD into those with normal and elevated SBPs. We found that the GWI, GWE, and GCW were lower and the GWW was higher in patients with CKD and normal SBP than in the CON group. This shows that when there is no afterload change, the work index clearly reflects the myocardial damage in patients with CKD. With the increase of the SBP and afterload, the GWI and GCW of patients with CKD increased significantly, which indicated that the MW of patients with CKD also increased with the increase of afterload, which was consistent with the findings of Chan et al.’s study [10]. There was no further decrease in the GLS in patients with CKD and elevated SBP, which was also affected by afterload. However, the GWW in patients with CKD increased further after the increase of the SBP load, indicating that the GWW can truly reflect the impairment of LV myocardial function and is not affected by afterload. In addition to the aforementioned reasons for myocardial asynchrony, the increase of the GWW may have been related to the increase of myocardial wall stress under higher afterload [10]. LV end-systolic hardness is an indicator of myocardial contractility, which reflects the ability of LV reverse pressure pumping at a higher level related to myocardial contractility enhancement, and the LV end-systolic hardness is higher in patients with hypertension than in those without hypertension [23]. At the same time, because GWE = (GCW − GWW)/GCW, GWE remained basically unchanged in patients with CKD and elevated SBP due to the almost proportional increase of the GCW and GWW.

### 4.4. A New Index for Detecting Myocardial Injury in Patients with Chronic Kidney Disease

The GLS is a predictor of adverse events, and it outweighs LVEF in this aspect [24,25]. However, the main limitation of the GLS is its load dependency [9,26,27]. Some studies have shown that PSL takes into account the impact of afterload, which can better reflect the myocardial work [10,20]. In the present study, we observed that the parameters of MW were superior to GLS in distinguishing myocardial injury between patients with CKD and healthy controls. The area under the ROC curve for the GWW was larger, regardless of the presence or absence of LVH and an elevated SBP. Therefore, the GWW can more sensitively detect myocardial injury in patients with CKD, and it can serve as a new index for noninvasive evaluation of myocardial function in patients with CKD.

### 4.5. Limitations

This present study has several limitations. First, in this study, the brachial artery systolic pressure measured by the cuff, instead of invasive measurement, was used to assess the left ventricular peak pressure. The results of the two methods are not completely consistent [20]. Therefore, the pressure-strain analysis has some limitations. However, this method has shown a good correlation with invasive measurement results in most previous studies [11,28,29,30]. In addition, myocardial work may vary during the day due to the fluctuating blood pressure. Second, there was no CKD staging among the study population. Patients with asymmetric left ventricular wall hypertrophy were included in this study, and the left ventricular volume of these patients may be overestimated or underestimated due to the geometric assumptions of the Simpson’s method. In addition, only CKD patients with LVEF >50% were included in the study, which may limit the generalizability of our findings in clinical use. Third, the study population excluded patients with clinically diagnosed coronary heart disease but included patients with CKD and other diseases, which may have a certain impact on our results. Fourth, since we only evaluated the global work of the LV myocardium in patients with CKD, we did not distinguish the work of each segment of the myocardium. Fifth, speckle-tracking echocardiography technology depends on the frame rate and image resolution, and unfortunately, 15 (12.5%) patients with CKD in our study had poor images and were excluded from the study. Finally, the number of included patients with CKD in each subgroup was relatively small, and there were some limitations in the statistical comparisons. A study with a larger number of cases is needed to further evaluate the clinical practicability and value of this new parameter of myocardial function. This study is cross-sectional, so the prognostic significance of the results is not clear and needs to be verified by further research.

## 5. Conclusions

Noninvasive PSL analysis of the left ventricle can be used to evaluate the global MW in patients with CKD, and this method appears to be superior to strain analysis in assessing myocardial injury in these patients. The study justified the role of GWW in the noninvasive assessment of myocardial function in patients with CKD.

## Figures and Tables

**Figure 1 diagnostics-12-00856-f001:**
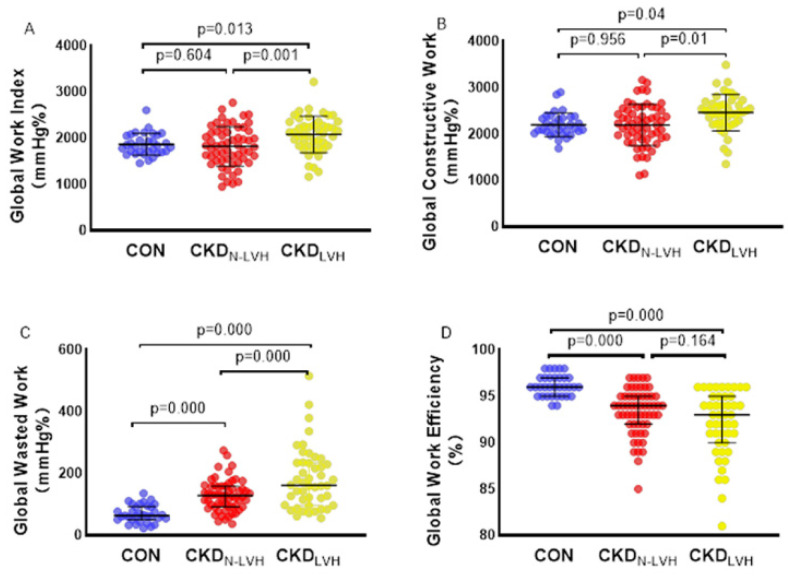
Effect of left ventricular hypertrophy on myocardial work. (**A**) Global work index, (**B**) global constructive work, (**C**) global waste work, (**D**) global work efficiency. CON: control group; CKD_N-LVH_: the chronic kidney disease with normal left ventricle group; CKD_LVH_: the chronic kidney disease with left ventricular hypertrophy group.

**Figure 2 diagnostics-12-00856-f002:**
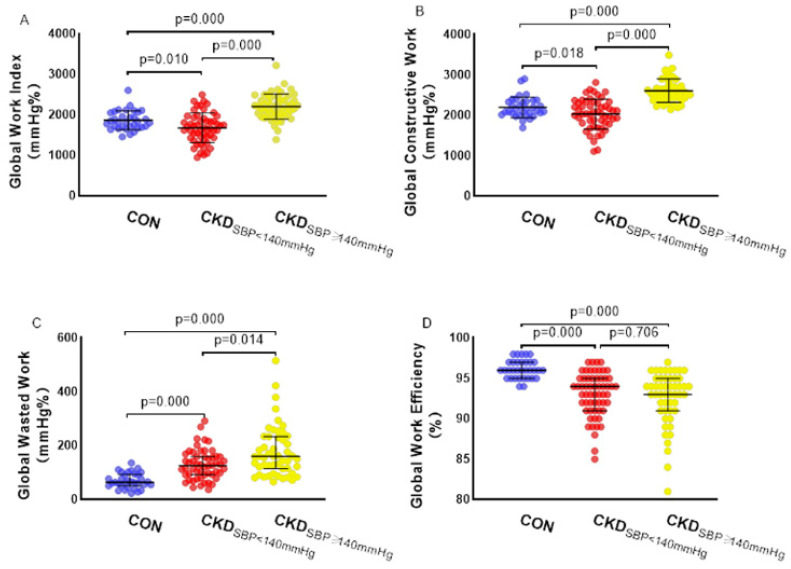
Effect of systolic blood pressure on myocardial work. (**A**) Global work index, (**B**) global constructive work, (**C**) global waste work, (**D**) global work efficiency. CON: control group; CKD_SBP__<140_
_mmHg_: chronic kidney disease with systolic blood pressure < 140 mmHg group; CKD_SBP__≥140_
_mmHg_: chronic kidney disease with systolic blood pressure ≥ 140 mmHg group.

**Figure 3 diagnostics-12-00856-f003:**
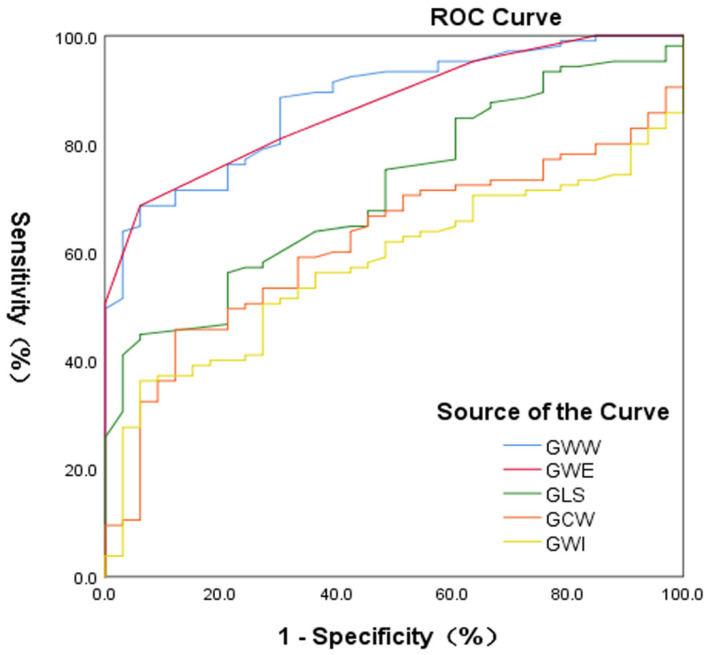
Receiver-operating characteristic (ROC) curve analyses for the accuracy of GWI, GCW, GWW, GWE, and GLS parameters to identify chronic kidney disease patients with myocardial injury. The analyses include all study participants (N = 138). GLS, global longitudinal strain; GWI, global work index; GCW, global constructive work; GWW, global waste work; GWE, global work efficiency.

**Figure 4 diagnostics-12-00856-f004:**
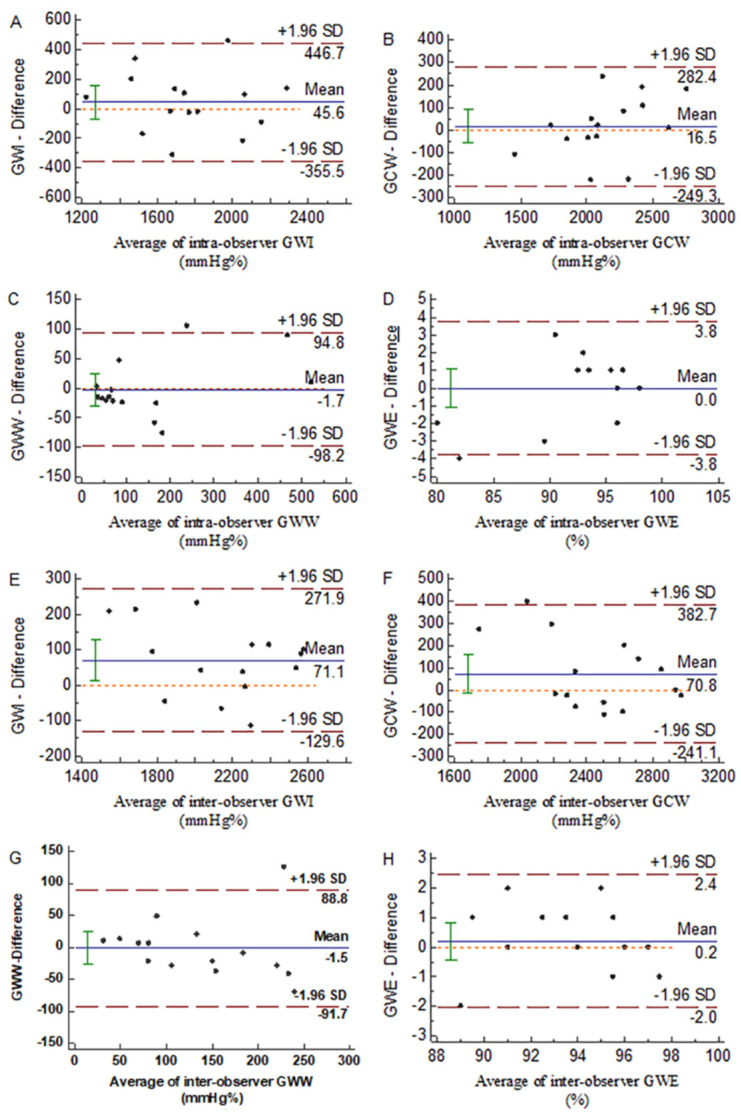
Bland−Altman plots of intra-observer agreement for: (**A**) GWI, (**B**) GCW, (**C**) GWW, and (**D**) GWE; Bland−Altman plots of inter-observer agreement for: (**E**) GWI, (**F**) GCW, (**G**) GWW, and (**H**) GWE. GWI, global work index; GCW, global constructive work; GWW, global waste work; GWE, global work efficiency.

**Table 1 diagnostics-12-00856-t001:** Clinical and laboratory characteristics.

	CON (*n* = 33)	CKD_N-LVH_ (*n* = 59)	CKD_LVH_ (*n* = 46)	*p*
Age (years)	48.1 ± 8.9	53.2 ± 16.1	54.4 ± 15.3	0.134
Male gender, *n* (%)	18 (54.5%)	39 (66.1%)	22 (47.8%)	0.161
SBP (mmHg)	119.8 ± 9.4	130.8 ± 17.5 *	150.9 ± 17.2 *†	0.000
DBP (mmHg)	78.6 ± 7.8	81.4 ± 10.5	87.7 ± 13.1 *†	0.001
BMI (kg/m^2^)	23.7 ± 2.1	22.9 ± 3.2	23.5 ± 3.5	0.459
BSA (m^2^)	1.7 ± 0.18	1.69 ± 0.17	1.65 ± 0.17	0.441
HR (bpm)	66.2 ± 8.8	73.6 ± 10.4 *	73.2 ± 10.5 *	0.002
eGFR (mL/min/1.73 m^2^)	104.5 ± 14.9	35.1 ± 27.8 *	15.4 ± 17.5 *†	0.000
SCR (umol/L)	69.5 ± 15.9	356.7 ± 340.2 *	596.7 ± 359.5 *†	0.000
BUN (mmol/L)	4.68 ± 1.14	14.67 ± 9.45 *	20.43 ± 11.91 *†	0.000

Data are expressed as mean ± SD or as number (percentage). CKD, chronic kidney disease; SBP, systolic blood pressure; DBP, diastolic blood pressure; BMI, body mass index; BSA, body surface area; HR, heart rate; eGFR, estimated glomerular filtration rate; SCR, serum creatinine; BUN, blood urea nitrogen. * *p* < 0.05 vs. CON. † *p* < 0.05 vs. CKD_N-LVH._

**Table 2 diagnostics-12-00856-t002:** Transthoracic echocardiographic parameters.

	CON (*n* = 33)	CKD_N-LVH_ (*n* = 59)	CKD_LVH_ (*n* = 46)	*p*
IVSd (mm)	9.24 ± 1.25	9.51 ± 1.33	11.7 ± 1.94 *†	0.000
LVIDd (mm)	46.55 ± 4.15	46.1 ± 4.02	50.26 ± 5.14 *†	0.000
PWTd (mm)	7.94 ± 1.12	9.01 ± 1.43	11.07 ± 1.82 *†	0.000
LVEDV (mL)	101.36 ± 21.61	98.97 ± 19.51	121.3 ± 28.52 *†	0.000
LVEDVI (mL/m^2^)	59.76 ± 10.35	58.6 ± 9.54	73.69 ± 15.43 *†	0.000
LVEF (%)	68.52 ± 6.85	69.37 ± 6.06	66 ± 7.26 †	0.036
FS (%)	38.58 ± 5.43	39.08 ± 4.88	36.87 ± 5.68	0.098
LVMI (g/m^2^)	78.7 ± 15.37	85.62 ± 15.56	133.99 ± 25.1 *†	0.000

Data are expressed as mean ± SD. CKD, chronic kidney disease; LVH, left ventricular hypertrophy; IVSd, interventricular septum diameter at end-diastole; LVIDd, left ventricular internal diameter at end-diastole; PWTd, posterior wall thickness at end-diastole; LVEDV, left ventricular end-diastolic volume; LVEDVI, left ventricular end-diastolic volume index; LVEF, left ventricular ejection fraction; FS, fraction shortening; LVMI, left ventricular mass index. * *p* < 0.05 vs. CON. † *p* < 0.05 vs. CKD_N-LVH_.

**Table 3 diagnostics-12-00856-t003:** Left ventricular global longitudinal strain and myocardial work analysis.

	CON (*n* = 33)	CKD_N-LVH_ (*n* = 59)	CKD_LVH_ (*n* = 46)	*p*
GLS (%)	20.05 ± 1.77	18.87 ± 2.16 *	17.82 ± 2.48 *†	0.000
PSD (%)	39.31 ± 7.94	50.7 ± 10.96 *	61.23 ± 13.85 *†	0.000
GWI (mmHg%)	1865.1 ± 235.3	1822.3 ± 427.9	2081.7 ± 393.5 *†	0.002
GCW (mmHg%)	2201.5 ± 256.2	2196.7 ± 445.3	2462.9 ± 391.4 *†	0.001
GWW (mmHg%)	64 (51, 92)	129 (92.5, 158.5) *	162 (98, 235) *†	0.000
GWE (%)	96 (95, 97)	94 (92, 95) *	93 (90, 95) *	0.000

Data are expressed as mean ± SD or as number (percentage). CKD, chronic kidney disease; GLS, global longitudinal strain; PSD, peak strain dispersion; GWI, global work index; GCW, global constructive work; GWW, global waste work; GWE, global work efficiency. * *p* < 0.05 vs. CON. † *p* < 0.05 vs. CKD_N-LVH_.

**Table 4 diagnostics-12-00856-t004:** Left ventricular global longitudinal strain and myocardial work according to the subdivision analysis of the SBP.

	GLS (%)	PSD (%)	GWI (mmHg%)	GCW (mmHg%)	GWW (mmHg%)	GWE (%)
CON (*n* = 33)	20 ± 1.8	39.3 ± 7.9	1865.1 ± 235.3	2201.5 ± 256.2	64 (51, 92)	96 (95, 97)
CKD (SBP < 140 mmHg, *n* = 54)	18.5 ± 2.5 *	50.7 ± 10.6 *	1679.6 ± 366.2 *	2033.5 ± 372.7 *	126 (92, 159) *	93.5 (91, 95) *
CKD (SBP ≥ 140 mmHg, *n* = 51)	18.3 ± 2.2 *	60.2 ± 14.3 *†	2207.5 ± 313.3 *†	2609.7 ± 288.3 *†	160 (115, 232) *†	93 (91, 95) *

Data are expressed as mean ± SD or as number (percentage). SBP, systolic blood pressure; CKD, chronic kidney disease; GLS, global longitudinal strain; PSD, peak strain dispersion; GWI, global work index; GCW, global constructive work; GWW, global waste work; GWE, global work efficiency. * *p* < 0.05 vs. CON. † *p* < 0.05 vs. CKD_SBP<140 mmHg_.

## Data Availability

The data presented in this study are available on request from the corresponding author.

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
