# Peer review of "Noninvasive Evaluation of Myocardial Work in Patients with Chronic Kidney Disease Using Left Ventricular Pressure-Strain Loop Analysis"

_diagnostics, 2022, doi:10.3390/diagnostics12040856_

Round 1
Reviewer 1 Report
1) it is not well understood how ischemic heart disease, severe valvular disease and severe arrhythmias were defined as basic exclusion criteria;
2) “The accuracy of GWE and GWW in the diagnosis of myocardial injury in patients with CKD was better than that of GLS. GWE's ROC analysis showed that the best cutoff value was ≤94 mmHg%, with a specificity of 93.9% and a sensitivity of 68.6% ”)
3) It should be specified why patients with LVEF <50% were excluded, since a correlation between “myocardial work” and myocardial injury was studied;
4) The authors specified that it is a prospective study, but no follow-up period has been declared. In addition, it seems more of a cross-sectional study, the echocardiographic values ​​being measured only once in the respective population, without finding other values ​​at a distance.
5) I believe that the investigated outcomes and the timing of the measurement over time should be described in the chapter on methods.
I found several studies on myocardial work in CKD:
https://pubmed.ncbi.nlm.nih.gov/33433746/
https://pubmed.ncbi.nlm.nih.gov/33084159/
https://www.sciencedirect.com/science/article/pii/S1056872721000581
https://pesquisa.bvsalud.org/portal/resource/pt/wpr-868055
Only one of them has an impact on outcomes
Author Response
Response to Reviewer 1 Comments
Point 1: it is not well understood how ischemic heart disease, severe valvular disease and severe arrhythmias were defined as basic exclusion criteria;
Response 1: Thank you very much for your reminding .The ischemic heart disease, severe valvular disease and severe arrhythmias were defined according to the diagnosis of physicians and the results of routine electrocardiogram and echocardiography.
Point 2: “The accuracy of GWE and GWW in the diagnosis of myocardial injury in patients with CKD was better than that of GLS. GWE's ROC analysis showed that the best cutoff value was ≤94 mmHg%, with a specificity of 93.9% and a sensitivity of 68.6% ”)
Response 2: Thank you for your kindly comments. It has been corrected as follows, which is highlighted in red on Page 8(L232).
“The accuracy of GWE and GWW in the diagnosis of myocardial injury in patients with CKD was better than that of GLS. GWE's ROC analysis showed that the best cutoff value was ≤94 %, with a specificity of 93.9% and a sensitivity of 68.6% .”
Point 3: It should be specified why patients with LVEF <50% were excluded, since a correlation between “myocardial work” and myocardial injury was studied;
Response 3: Thank you very much for your instructive advice. Many previous studies have been proved that strain was a good index for predicting cardiac events. However, the main limitation of the parameter is its load dependency. An increase in afterload may lead to a lower strain value, which leads to the misjudgment of the real contractile function. Some studies have shown that PSL takes into account the impact of afterload, which can better reflect the myocardial work.
The study was designed to detect the early manifestations of myocardial injury in patients with chronic kidney disease, that is, to examine the sensitivity of myocardial work in dectecting early myocardial injuries in patients without the manifestation of reduced contractile function, so the patients with LVEF < 50% are excluded. Thank you very much for your valuable advice, we will conclude the patients with EF < 50% in our further studies.
1.Chan J, Shiino K, Obonyo NG, Hanna J, Chamberlain R, Small A. Left ventricular global strain analysis by two-dimensional speckle-tracking echocardiography: thelearning curve. J Am Soc Echocardiogr 2017;30:1081–90.
2.Mignot A, Donal E, Zaroui A, Reant P, Salem A, Hamon C. Global longitudinal strain as a major predictor of cardiac events in patients with depressed left ventricular function: a multicenter study. J Am Soc Echocardiogr 2010;23:1019–24.
3.Cho GY, Marwick TH, Kim HS, Kim MK, Hong KS, Oh DJ. Global 2-dimensional strain as a new prognosticator in patients with heart failure. J Am Coll Cardiol 2009;54:618–24.
4.Hubert A, Le Rolle V, Leclercq C, Galli E, Samset E, Casset C et al. Estimation of myocardial work from pressure-strain loops analysis: an experimental evaluation. Eur Heart J Cardiovasc Imaging 2018; doi:10.1093/ehjci/jey024.
5.Chan, J.; Edwards, N.F.A.; Khandheria, B.K.; Shiino, K.; Sabapathy, S.; Anderson, B.; Chamberlain, R.; Scalia, G.M. A new approach to assess myocardial work by non-invasive left ventricular pressure-strain relations in hypertension and dilated cardiomyopathy. Eur. Heart J. Cardiovasc. Imaging 2019, 20, 31–39, doi:10.1093/ehjci/jey131.
Point 4: The authors specified that it is a prospective study, but no follow-up period has been declared. In addition, it seems more of a cross-sectional study, the echocardiographic values ​​being measured only once in the respective population, without finding other values ​​at a distance.
Thanks for your suggestion. The study was initially designed as a prospective research with following meaurements over time. During the research, our team did make follow-ups, but the loss rate of the follow-up was relatively high due to the influence of COVID-19. So far, we didn’t collect enough following-up data for analysis. Since the lack of following-up data, I agree to define it as a cross-sectional study.
Point 5: I believe that the investigated outcomes and the timing of the measurement over time should be described in the chapter on methods.
Response 5: The authors feel sorry for the confusion. The study was designed as a prospective research with following meaurements over time. However, due to the continuous influence of COVID-19, the loss rate of the follow-up was so high that we cannot collect enough data for analysis. I will add this aspect to my limitations: This study is cross-sectional study, and the prognostic significance of the results is not clear and needs to be verified by further research.

Reviewer 2 Report
The manuscript of “Noninvasive Evaluation of Myocardial Work in Patients with Chronic Kidney Disease Using Left Ventricular Pressure-Strain Loop Analysis” by Xiaohua Liu and co-authors aims to evaluate the function of the left ventricle in patients with chronic kidney disease (CKD) by the new method speckle-tracking echocardiography-based left ventricular pressure-strain loop analysis (LV PSL analysis). The authors concluded that the global waste work (GWW) index can more sensitively detect myocardial injury in patients with CKD, and it can serve as a new index for noninvasive evaluation of myocardial function in patients with CKD. The manuscript is well structured and written; the conclusions are supported by the analysis of the results presented; and therefore, the manuscript may be accepted for publication after minor revision.
Comments:
- Figs. 1 and 2: Group names (especially subscripts) on the x-axis are hard to see. The groups CKDLVH and CKDN-LVH should be fully deciphered in the legends to the figures.
- Section 3.6. Accuracy of Identification of Myocardial Injury: In the text, the results are presented as percentage changes, while the specificity and sensitivity data in Figure 3 are presented in conventional units from 0 to 1. Please, explain.
- Please, add the decoding of the abbreviation GLS to the legend.
- In the Statistical Analysis Section, it is necessary to specify the method for assessing normality.
- The Conclusion section contains rather general statements about the LV PSL analysis, which have already been made by other authors (DOI: 10.1007/s10554-020-02132-9). The section could be improved and more specific. For example, one could justify the calculation of the GWW index for non-invasive assessment of myocardial function in patients with CKD.
- The Limitations section: More limitations could be mentioned. For example, stages of CKD, the need for geometric assumptions when calculating LV volumes in patients with asymmetric LV remodeling, etc.
- The References section contains only 24 references and can be expanded by adding recent reviews about the advantages and disadvantages of LV PSL analysis (For example: https://doi.org/10.1007/s10741-021-10119-4, or others).
Round 2
Reviewer 1 Report
please provide a professional english proofread certificate
no further comments
